# Structural basis of mitochondrial translation

**Shintaro Aibara[1†], Vivek Singh[1,2], Angelika Modelska[3‡], Alexey Amunts[1,2]***

[1]Science for Life Laboratory, Department of Biochemistry and Biophysics, Stockholm University, Solna, Sweden; [2]Department of Medical Biochemistry and Biophysics, Karolinska Institutet, Stockholm, Sweden; [3]Laboratory of Translational Genomics, Centre for Integrative Biology, University of Trento, Trento, Italy

**Abstract** Translation of mitochondrial messenger RNA (mt-mRNA) is performed by distinct mitoribosomes comprising at least 36 mitochondria-specific proteins. How these mitoribosomal proteins assist in the binding of mt-mRNA and to what extent they are involved in the translocation of transfer RNA (mt-tRNA) is unclear. To visualize the process of translation in human mitochondria, we report ~3.0 Å resolution structure of the human mitoribosome, including the L7/L12 stalk, and eight structures of its functional complexes with mt-mRNA, mt-tRNAs, recycling factor and additional trans factors. The study reveals a transacting protein module LRPPRC-SLIRP that delivers mt-mRNA to the mitoribosomal small subunit through a dedicated platform formed by the mitochondria-specific protein mS39. Mitoribosomal proteins of the large subunit mL40, mL48, and mL64 coordinate translocation of mt-tRNA. The comparison between those structures shows dynamic interactions between the mitoribosome and its ligands, suggesting a sequential mechanism of conformational changes.

**\*For correspondence:**
amunts@scilifelab.se

**Present address:** [†]Department of Molecular Biology, Max-Planck-Institute for BiophysicalChemistry, Göttingen, Germany; [‡]Aix Marseille Université, CNRS, INSERM, Centre d'Immunologie deMarseille-Luminy (CIML), Marseille, France

**Competing interests:** The authors declare that no competing interests exist.

## Introduction

Translation in humans takes place in the cytosol and mitochondria. Mitochondrial translation is responsible for the maintenance of the cellular energetic balance through synthesis of proteins involved in oxidative phosphorylation. This is required for adenosine triphosphate (ATP) production and the folding of the cristae. Therefore, impaired mitochondrial translation results in severe combined respiratory chain dysfunction leading to diminished ATP production and consequent cellular energy deficit. This condition is pathogenic in humans, causing myopathies and neurodegenerative diseases (*Boczonadi and Horvath, 2014*). The defects can happen at any age on any organs. For example, a recent study demonstrated that with age, decreased mitochondrial protein synthesis lead to adult-onset obesity that results in liver steatosis and cardiac hypertrophy (*Perks et al., 2017*). Therefore, changes in mitochondrial gene expression have long-term consequences on energy metabolism, and surveys suggest that mitochondrial diseases affect 2–5 in 10,000 individuals, mostly occurring due to disrupted mitochondrial gene expression (*Chinnery, 2014*).

Translation in mitochondria is carried out by specialized mitoribosomes that are structurally distinct with 2-fold reduced mt-rRNA and ~1 MDa increased protein mass. Specific autosomal recessive mutations associated with these extra proteins cause severe, infantile onset disease with growth retardation, neurological phenotypes and cardiac involvement, and are associated with Perrault syndrome (*De Silva et al., 2015*; *D'Souza and Minczuk, 2018*; *Boczonadi et al., 2018*; *Gardeitchik et al., 2018*; *Bugiardini et al., 2019*). Analysis of individuals with variations in mitoribosomal proteins demonstrated that mutations result in Leigh syndrome (*Lake et al., 2017*). Oxidative phosphorylation enzymology performed in skeletal muscle and liver cell lines from the patients showed deficiency of one or more complexes in which components are synthesized by the mitoribosomes. Further underscoring the importance of mitoribosomes in the regulation of cellular energy

metabolism, the expression of its components was also found to be modified in numerous cancers, a trait that has been linked to tumorigenesis and metastasis (*Kim et al., 2017*). Taken together, these pathomechanisms involve different stages of the mitoribosome activities, which have not been thoroughly investigated. Although advanced approaches to visualize mitochondrial processes have been developed (*Wallis et al., 2020*), the basic molecular mechanism of how human mitoribosome orchestrates the flow of genetic information from mt-DNA encoded genes to functional proteins is yet to be characterized. Since recent work demonstrates that employment of cell signalling molecules and networks has a potential of targeted therapies for mitochondrial diseases (*Ferreira et al., 2019*), the basic understanding of the underlying molecular mechanisms is crucial.

On the mechanistic level, upon transcription and maturation the flow of genetic information in mitochondria requires mt-mRNAs coding for 13 proteins to be delivered to the mitoribosome. Human mt-mRNAs have unusual features, and they lack conventional 5' and 3' untranslated regions, Shine-Delgarno sequences and 5' 7-methylguanosine caps. Consequently, specific RNA-binding proteins were identified that complement the unusual features of mt-mRNAs (*Rackham et al., 2012*). However, despite emerging evidence of the importance of RNA-binding proteins to translation, no mechanism for directly modulating the mitoribosome has been illustrated.

During translation on the mitoribosome, mt-tRNAs use anticodons to decode the mt-mRNA, and acceptor terminus (CCA 3') that is esterified to a cognate amino acid. Conventionally, the structural recognition between a tRNA and a ribosome is based on the canonical L-shape of tRNAs, where D- and T-loops interact with each other, giving rise to the elbow-like structure. This is followed by the transit of tRNA through the three distinct sites on the mitoribosome: aminoacyl (A-site), peptide (P-site), and exit (E-site). However, in human mitochondria, the genetic arrangement and the structures of human mt-tRNAs are remarkably diverse, including non-canonical and truncated species with reduced D- and/or T-loops (*Helm et al., 2000*; *Salinas-Giegé et al., 2015*). How the rearranged domains of human mt-tRNA are encased by the mitoribosomal elements and which contacts stabilize and translocate mt-tRNAs to perform translation is currently not known.

Progress has been made in obtaining structural data on the mitoribosomes from porcine tissues (*Greber et al., 2014*; *Greber et al., 2015*), HEK293S-derived cells (*Brown et al., 2014*; *Amunts et al., 2015*; *Brown et al., 2017*), and initiation complexes have been reconstituted (*Kummer et al., 2018*; *Khawaja et al., 2020*). While these data yielded the first snapshots, the available structural models are incomplete with key components responsible for mt-mRNA and mt-tRNA binding missing, which reflects the dynamic nature of translation. The structure of human mitoribosome at 3.5 Å resolution revealed an overall diverged architecture with 36 mitochondria-specific proteins (*Amunts et al., 2015*), suggesting that it would have functional characteristics distinguishing mitochondrial translation from the counterpart systems. This is further supported by the unique non-ribosomal features of translation in mitochondria, including leaderless mt-mRNAs, unusual structural characteristics of mt-tRNAs and an abundance of auxiliary factors (*Ott et al., 2016*). Therefore, a high-resolution structure of the actively translating human mitoribosome is needed to discuss the mechanism of action and functional contribution of those factors.

To provide insight into the cooperation between the mitochondria-specific features, newly identified components and mitoribosomal ligands, as well as to reveal how the protein synthesis function in human mitochondria is modulated by mitoribosomal proteins, we determined eight cryo-EM structures of human mitoribosomes in complex with mt-mRNA, mt-tRNAs and additional factors in different states. These structures report on sequential steps for mt-mRNA binding, mt-tRNA movement, and identify key components for translation in human mitochondria.

## Results

### Structure determination and improved model of human mitoribosome

We reasoned that a suitable source for structural studies would be mitochondria from dividing cells grown in nutrient-rich expression medium requiring active translation. Mitoribosomal complexes were stalled with antibiotics quinupristin and dalfopristin that bind to the exit tunnel and peptidyl-transferase center, respectively (*Harms et al., 2004*; *Noeske et al., 2014*). Cryo-EM imaging of these mitoribosomes resulted in an overall ~3.0 Å resolution map from 143,851 particles (*Figure 1—figure supplement 1*, *Figure 1—figure supplement 2*; *Supplementary file 1*). This enabled us to

build the model of the human mitoribosome, including specific elements responsible for ligand binding, and the L7/L12 stalk (*Figure 1*; *Supplementary file 2*).

Six copies of bL12m N-terminal domain, including loops, are resolved in this reconstruction, as well as the improved model for mL54 was assigned on the basis of the new densities (*Figure 1B,C*, *Figure 1—figure supplement 3*, *Figure 1—figure supplement 4*). In addition to the N-terminal extension of uL10 that stabilizes the L7/L12 stalk, our map also reveals N-terminal mitochondrial extension of bL12m (*Nouws et al., 2016*), linking it to the mitoribosomal core. Sixteen residues could be assigned (0 to −15), and contacts with mL53 and uL10m identified, so that Leu-13 protrudes into a hydrophobic patch on uL10m (*Figure 1B*). In the mitochondria-specific protein mL54, a 19-residue loop Ala72-Ala90 is connected to the hydrophobic pocket of uL11m extension with Ile77 and Tyr78 (*Figure 1C*). Together, these newly modeled components explain how the functional L7/L12 stalk of the human mitoribosome is stabilized.

Since weak but distinct densities in the tRNA binding sites were observed in the consensus map, we performed 3D classification employing signal subtraction (*Scheres, 2016*) on those regions in a sequential manner (*Figure 1—figure supplement 1*). This yielded eight structurally homogeneous classes corresponding to different states of the translation cycle with mt-tRNAs. Mitoribosomes with the complete set of mt-tRNAs occupying A-site, P-site, and E-site are represented by 9% of the particles; A- and P-sites are occupied in 10%; P- and E-sites are occupied in 17%; P-site only is occupied in 5%; E-site only is occupied in 25%. In addition, we were able to depict two intermediate states: 1) intersubunit A/P- and P/E-hybrid state in 3%, wherein mt-tRNA in the A-site is rotated towards the exit tunnel, and mt-tRNA in the P-site is tilted towards the E-site, coupled with subunit rotation; 2) intersubunit P/E-hybrid in 14% (*Figure 1—figure supplement 1*, *Figure 1—figure supplement 2*). Together, the ensemble of reconstructions includes an an unrotated, rotated and a post-translocation state, indicative of a sequential mt-tRNA movement along translation.

In a subset of particles, a density was found in the A-site with a recognizable three-helix bundle corresponding to domain I of the native mitochondrial ribosome recycling factor (mt-RRF) (*Figure 1—figure supplement 1* bottom-left panel, *Figure 1—figure supplement 5*). The mitoribosomes in this class adopt a ratcheted state, with the small subunit exhibiting 7.5° rotation. The comparison with the recently reconstituted complex of mt-RRF (*Koripella et al., 2019*) shows that the conformation of the head is different, as well as the interactions with mt-RRF (*Figure 1—figure supplement 5*). Particularly, the mitochondria-specific N-terminal extension does not adopt a helical conformation as previously suggested (*Koripella et al., 2019*), and reported interactions with rRNA are not observed in our native complex.

## Mitochondria-specific elements involved in mt-tRNA binding

The captured mt-tRNAs allowed us to identify key mitochondria-specific elements responsible for their anchoring to each one of the binding sites on the translationally active mitoribosome. In the A-site, in addition to the bacteria-like interactions of uL16m and H89 rRNA, the mitochondria-specific C-terminal helix of uL11m extends towards the mt-tRNA elbow, while the conventional H38 'A-site finger' is missing, and no bacteria-like interactions of uL27m with A-tRNA are detected (*Voorhees et al., 2009*). Instead, a specific proteinaceous moiety interacts with the A-tRNA elbow. It is formed by the mitochondria-specific proteins mL40 and mL48 with their N-terminal and C-terminal helices, respectively (*Figure 2*, *Figure 2—figure supplement 1*). Both of these proteins are anchored in the central protuberance, through extensive interactions with the structural tRNA$^{Val}$ (CP-tRNA), as well as shared with bL31m and interactions with mitochondria-specific protein mL46 (*Figure 2*, *Figure 2—figure supplement 1*). In the tRNA binding site, the N-terminal helix of mL40 with residues Lys66, Lys70, and Gln63 approach the elbow region of the A-tRNA (*Figure 2—figure supplement 2A*). The C-terminal helix of mL48 does not directly contact mt-tRNA, but supports the N-terminal helix of mL40 via a hydrophobic interface, allowing it to contact mt-tRNA elbow in the A-site (*Figure 2—figure supplement 2B*).

The P-site has been remodeled substantially compared to bacteria as well. Consistent with the previous data, the positively charged C-terminal tail of uS9 (Lys395) approaches the P-site tRNA anticodon stem-loop (*Figure 3—figure supplement 1*; *Selmer et al., 2006*; *Greber et al., 2015*; *Kummer et al., 2018*). This recognition is crucial for translation fidelity (*Arora et al., 2013*; *Jindal et al., 2019*). The N-terminal helix of mL40 that binds A-tRNA also interacts with P-tRNA elbow. A series of positively charged residues, Arg67, Arg70, and Lys77 interact with the phosphate

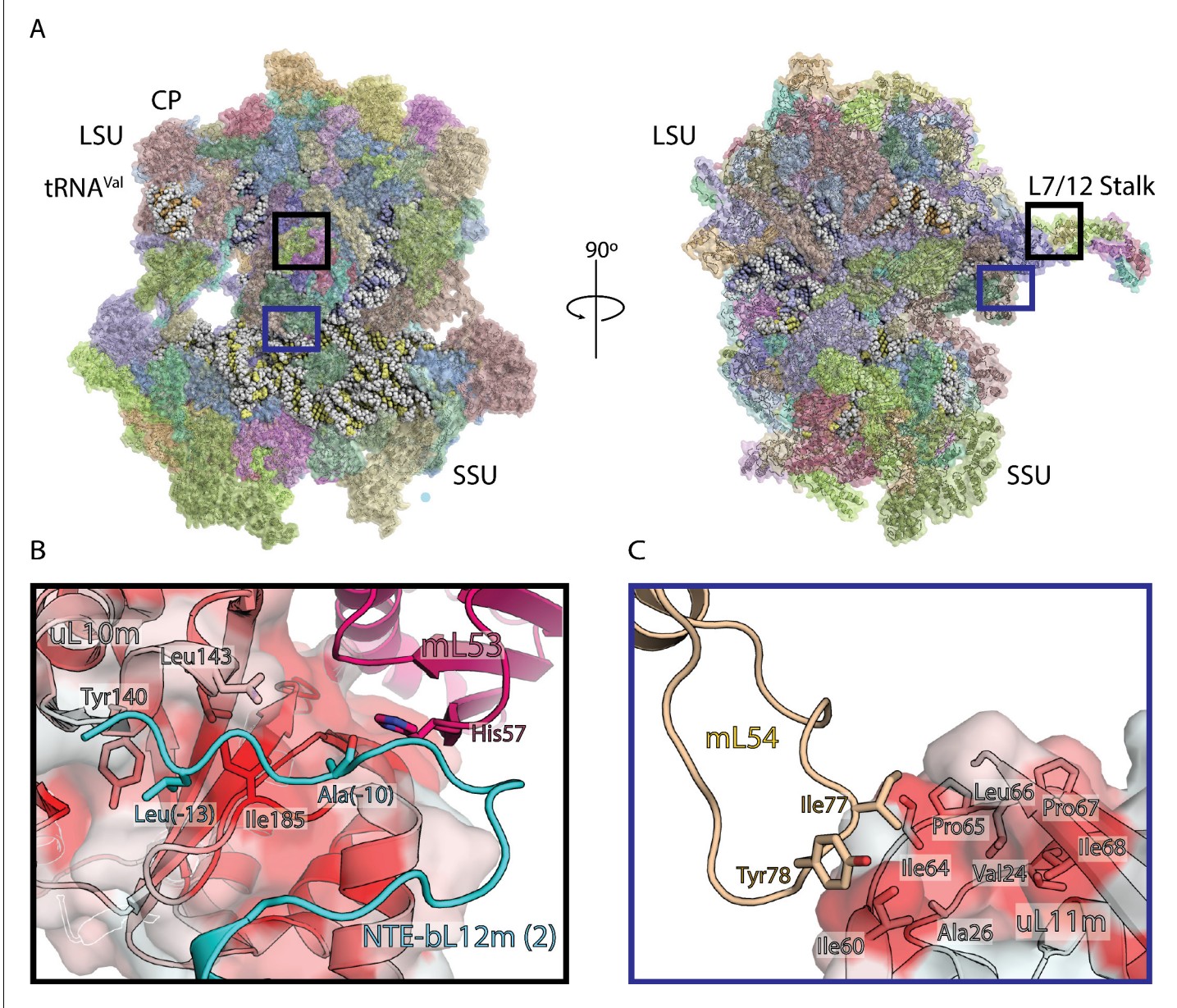

**Figure 1.** Structure of human mitoribosome and modeled L7/L12 stalk. (**A**) Overview of the human mitoribosome model. The mitoribosomal proteins are shown in cartoon with transparent surface, rRNA is shown as spheres. Zoomed in regions are indicated. (**B**) View of the improved model of the N-terminal mitochondrial extension of bL12m (cyan) and its contacts with mL53 (magenta) and uL10m (grey). The protein uL10m is shown with transparent surface colored by hydrophobicity, with red indicating most hydrophobic, revealing a hydrophobic patch involved in bL12m binding. (**C**) The modeled loop of mL54 (wheat) forms hydrophobic interface with uL11m (grey with transparent surface colored by hydrophobicity). These interactions between mitochondria-specific elements contribute to the stability of the stalk.

The online version of this article includes the following figure supplement(s) for figure 1:

**Figure supplement 1.** Cryo-EM data processing.
**Figure supplement 2.** Fourier shell correlation (FSC) curves for mitoribosomal complexes.
**Figure supplement 3.** Map and model of human mitoribosome L7/L12 stalk.
**Figure supplement 4.** Map and model human mitoribosome mL54.
**Figure supplement 5.** Map and model human mt-RRF.

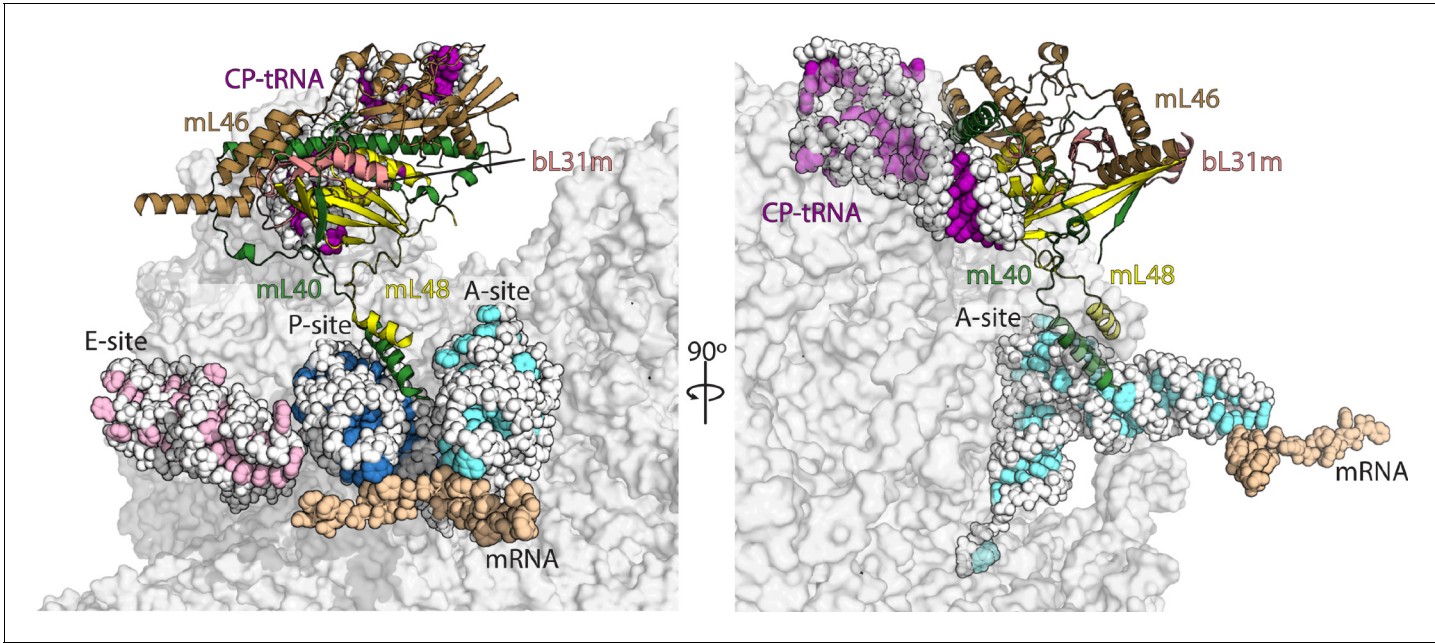

**Figure 2.** Interactions of mL40-mL48 in the tRNA binding sites and the central protuberance. Views of the tRNA binding sites of human mitoribosome showing interactions of mL40 (green) and mL48 (yellow) with A-tRNA (white/aquamarine) and P-tRNA (white/blue). In the central protuberance, mL40 and mL48 interact with mL46 (brown), bL31m (salmon) and CP-tRNA (white/pink). In the right panel, P- and E-tRNA are removed for clarity.

The online version of this article includes the following figure supplement(s) for figure 2:

**Figure supplement 1.** Topology and interactions of mL40 and mL48 in the central protuberance and with tRNAs.

**Figure supplement 2.** Hydrophobic interface between mL40 and mL48 involved in tRNA interactions.

backbone of the P-tRNA elbow (*Figure 2—figure supplement 2A*). We also observe that the uL27m N-terminal loop is stably associated with the acceptor arm of P-tRNA (*Figure 3—figure supplement 2*).

In the E-site, the mt-tRNA elbow is stabilized by the C-terminal helix of mL64, extending from the central protuberance, which was not previously resolved. E-tRNA is further stabilized by uS7m with its C-terminal helix aligned with the major groove of E-tRNA anticodon stem. The residues Lys221 and Lys228 form potential interactions with E-tRNA anti-codon stem backbone phosphates (*Figure 3—figure supplement 1*, *Figure 3—figure supplement 2*, *Video 1*). Together, the interactions of the mitochondria-specific proteins with mt-tRNAs suggest the existence of distinct tRNA translocation intermediates, which we have further analyzed below.

The mitoribosomes bound to A- and P-tRNA also contain additional density associated with the anticodon stem of A-tRNA, stretching from residue 40 down to the backbone phosphate of the last residue of mRNA codon in the A-site. The shape of the elongated 15 Å long density matches structures of polyamines, which are known to reside in mitochondria (*Tassani et al., 1996*). Polyamines are linear aliphatic hydrocarbons with at least three amino groups, and since four amino groups fit the density, it could be modeled it as spermine (*Figure 3—figure supplement 3*). Although we cannot explicitly state that the densities are indicative of the presence of spermine and rule out the possibility that an unidentified protein moiety is involved, the stable association with the A-tRNA containing classes suggests that the binding is consistent and

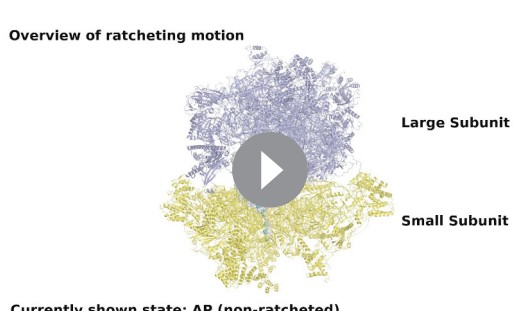

**Video 1.** Movie highlighting the protein elements that differ across tRNA states observed in this data. https://elifesciences.org/articles/58362#video1

functional, possibly bridging between the translocated mt-tRNAs.

## Conformational trajectory of mt-tRNA movement

We next examined which conformational changes of the mitoribosome occur by comparing individual structures. We monitor those changes from the perspective of the tRNA moving in a step-wise manner through the mitoribosome from A- to E-site over a distance of ~50 Å (*Figure 3*).

The binding of mt-tRNA to the A-site is held via interactions of the mt-tRNA elbow with the mitochondria-specific protein mL40. Upon the displacement of the acceptor arm from the A-site to A/P transition state, the mitoribosomal small subunit ratchets 6.5° and the head swivels. The following conformational changes, which are related to the formation of the hybrid state, allow accommodation of a rotated mt-tRNA. They involve primarily the N-terminal helix of mL40 that shifts 5 Å towards the P-site. Therefore, the protein element maintains the contact with mt-tRNA from A-site to A/P state, contributing to the stability of the transition state (*Figure 3B*). Upon the next transition from A/P state to the P-site, the mL40 N-terminal helix remains in contact with the mt-tRNA elbow through electrostatic interactions (*Figure 3C*, *Figure 2—figure supplement 2A*).

The movement of P-tRNA to the P/E state upon ribosome ratcheting is facilitated by the exchange of protein contacts from mL40 to mL64 (*Figure 3D*). Therefore, the elbow remains stabilized by the mitochondria-specific protein elements throughout the moving (*Figure 3—figure supplement 2*, *Video 1*). Our structures suggest that the released mL40 N-terminal helix plays a role in posing a steric barrier for a back transition to the P-site (*Figure 3C*).

Next, P/E to E-site movement is facilitated by the rotation-associated 6 Å conformational change of the mL64 C-terminal helix (*Figure 3E*, *Video 1*). While the overall contact is maintained, the specific contact points shift further along the helix towards the C-terminus, sliding the mt-tRNA elbow to the E-site. This suggests an active involvement of mL64 in the mt-tRNA movement. Further, the acceptor arm of the deacetylated mt-tRNA is stabilized by interactions with bL33m during the transition. Also here, the contact point of the ß2-ß3 loop (Arg36) and the N-terminal backbone amide of Ser65 is shifted to Lys61 in the ß4-ß5 loop (*Figure 3E*, *Figure 3—figure supplement 1*).

In regard to the L1 stalk, in the density maps we observe its binding to the elbow of the deacetylated tRNA, as the mitoribosome adopts the non-ratcheted state (from P/E to E-site) (*Figure 3—figure supplement 4*). This movement is defined as outward swinging (*Selmer et al., 2006*). Finally, the release from the mitoribosome is facilitated by the loss of interactions with E-tRNA upon ratcheting, that includes the conformational change of the mL64 C-terminal helix and the movement of uS7m away from the E-site (*Figure 3E*). This loss of interactions effectively destabilizes the tRNA on the mitoribosome.

## LRPPRC-SLIRP module delivers mRNA for translation by mitoribosome

The presence of mt-tRNA in A- or P-site of translating mitoribosomes correlates with a continuous density extending from the mitoribosomal pentatricopeptide repeat (PPR) protein mS39 to the decoding center. Comparison with the structures of the mitoribosome without P-site tRNA suggests that this density represents mt-mRNA (*Figure 3—figure supplement 5*). Furthermore, particles with bound mt-mRNA also exhibit an additional elongated density upstream of a non-mitoribosomal protein interacting with mS39 (*Figure 3—figure supplement 6*). This elongated density bound to the solvent side of mS39 embraces the bottom domain of the superhelical spiral shell and tightly interacts with at least ten tandem repeats. The two moieties share a surface area of ~2000 Å$^2$, representing by far the largest protein-protein interface on the mitoribosome. The upper globular domains of the new density and mS39 are separated from each other forming an overall heterodimer-like arrangement with two parted volute architectures pointing away from each other. While the mS39 volute is docked to the head of the small subunit, the new density counterpart extends away from the mitoribosome. Apart from mS39, no other direct interaction between the newly identified density and the mitoribosome is observed in our map. An in situ electron cryo-tomography study suggested presence of a moiety in a similar position (*Englmeier et al., 2017*), confirming that it is associated with active translation (*Figure 3—figure supplement 6*).

To identify the additional factor from the cryo-EM map, we performed focused 3D-refinement in attempt to improve the local resolution in this peripheral region. However, due to the intrinsically dynamic arrangement of the outer bound protein moiety, it was not possible to resolve the

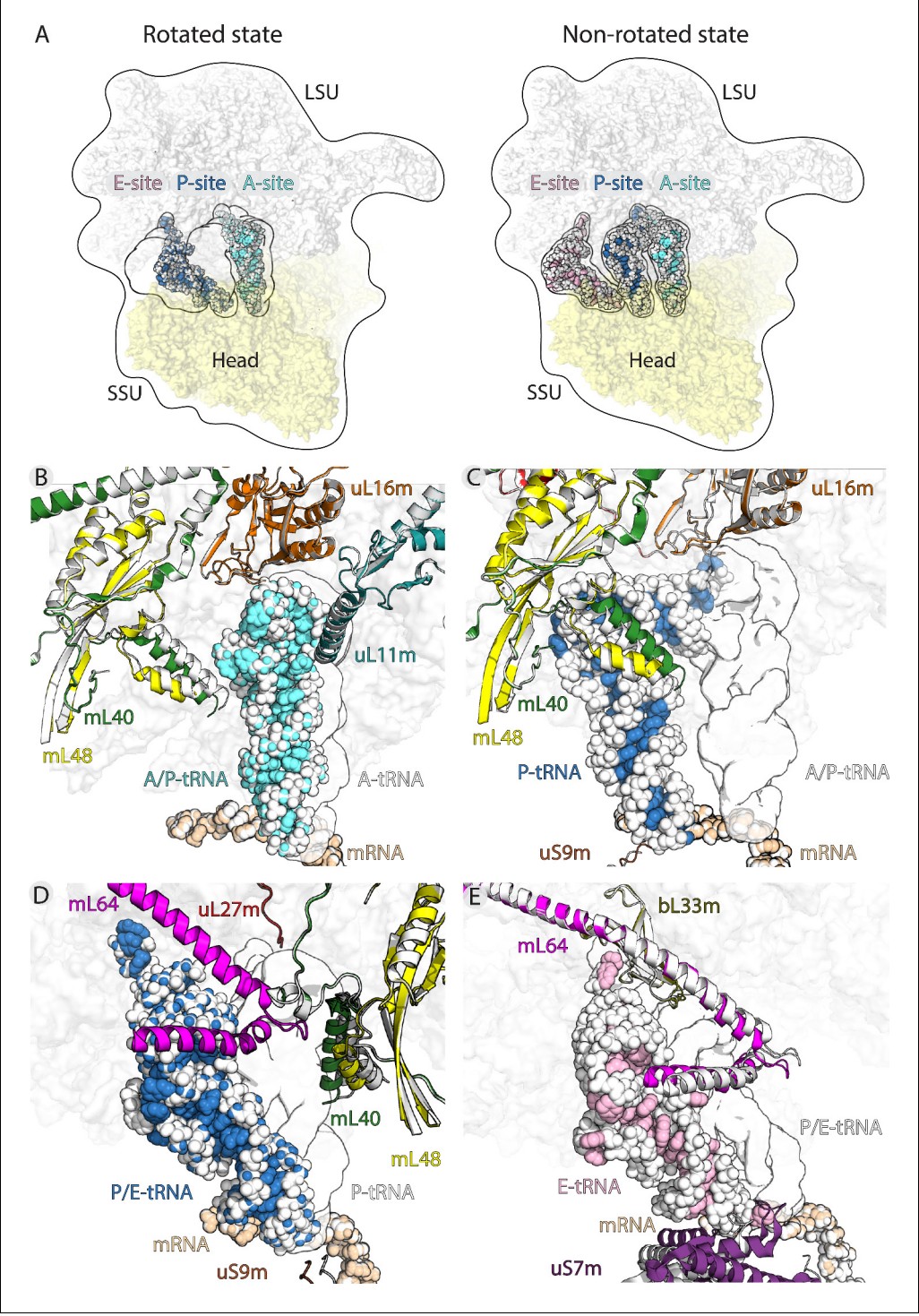

**Figure 3.** Translocation of mt-tRNA and involvement of mitoribosomal proteins. (**A**) Overview of tRNAs bound to the A-, P- and E-site in the rotated and non-rotated mitoribosome. The models were superimposed using the mt-LSU as a reference The classical tRNA binding sites are outlined. (**B**) Comparison between mt-tRNA in the A-site (outlined) and A/P state (light blue/white). (**C**) Comparison between mt-tRNA in the A/P state (outlined) and P-site (blue/white). (**D**) Comparison between mt-tRNA in the P-site (outlined) and P/E state (blue/white). (**E**) Comparison between mt-tRNA in the P/E state (outlined) and E-site (pink/white).

The online version of this article includes the following figure supplement(s) for figure 3:

**Figure supplement 1.** Interactions of mitoribosomal proteins with tRNAs.

*Figure 3 continued on next page*

**Figure supplement 2.** Interactions between mitoribosomal proteins and tRNA.
**Figure supplement 3.** Unmodeled density between A-site and P-site tRNAs.
**Figure supplement 4.** L1 stalk movement in ratcheted and non-ratcheted states.
**Figure supplement 5.** Density of mRNA in P-site tRNA bound mitoribosome.
**Figure supplement 6.** Density of LRPPRC-SLIRP complex in mRNA-bound mitoribosome.

secondary structures by cryo-EM. Therefore, we analyzed the highly purified native sample by mass spectrometry. Consistently with the size of the density, the analysis suggested leucine-rich PPR motif-containing protein (LRPPRC) (*Supplementary file 3*, *Supplementary file 4*). LRPPRC is conserved from flies to mammals and has a broad and strong RNA binding capacity through multiple PPR motifs (*Baggio et al., 2014*). Functionally interacting PPR proteins is a common theme in mitochondrial gene expression systems, and PPR heterodimers were previously reported in RNA metabolism systems (*Okuda and Shikanai, 2012*; *Aphasizhev and Aphasizheva, 2013*).

Additional studies also showed that LRPPRC forms a stable complex with an 11 kDa protein named Stem-Loop Interacting RNA binding Protein (SLIRP) (*Spåhr et al., 2016*). In vivo studies on mice liver mitochondria, including knockout, complemented by RNA sequencing of isolated mitoribosomes and the associated mt-mRNA showed that SLIRP is important for the presentation of mt-mRNA to the mitoribosome for efficient translation (*Lagouge et al., 2015*). Despite the small molecular weight, our mass spectrometry data validated the presence of SLIRP in the purified translating mitoribosomal complex that was used for cryo-EM (*Supplementary file 3*, *Supplementary file 4*). The quantified weighted spectra for SLIRP is similar to the mitoribosomal proteins, and 46% of the sequence has been covered through four peptides identified by the mass spectrometry experiment (*Supplementary file 4*). Furthermore, the shape of this moiety within the density map is consistent with the LRPPRC-SLIRP complex formed in vitro, which was also shown to have an elongated shape, as evident by a modified migration on a native gel (*Spåhr et al., 2016*). Finally, LRPPRC-SLIRP has not been detected in the control experiments with non-translating human mitoribosomes (*Amunts et al., 2015*). Therefore, translating mitoribosomes with mt-tRNA in the P-site and mt-mRNA are associated with the LRPPRC-SLIRP complex, and the contacts are driven by PPR-mediated interactions via mS39 (*Figure 4*).

## Discussion

In this study, the preparation of mitoribosomes from HEK cells led to the isolation of functional complexes with associated *trans*-acting factors. The structural analysis of translating complexes provided an improved model for the human mitoribosome, including the L7/L12 stalk, and revealed new insights into the functionally important regions involved in mt-tRNA translocation and mt-mRNA binding. Although less stable and short-lived intermediates of partially rotated states that involve minor conformational changes are likely to have escaped detection in our study, we identified populations corresponding to stable classical and hybrid states covering the entire span of mt-tRNA movement on the mitoribosome. The comparison between the states allowed describing a sequence of concerted steps in the mitochondrial translation that have not been previously reported.

The structural analysis revealed how the diverged mt-tRNAs fit in the protein-rich interface of the two mitoribosomal subunits, and how its trajectory is correlated with mitochondria-specific protein elements. Upon binding to the A-site and the recognition of the anticodon stem-loop, the tRNA elbow is stabilized by the mL40 N-terminal helix, while the mitochondria-specific extension of uL11m restricts a potential backward movement of the elbow. N-terminal helix of mL40 through electrostatic interactions further regulates the subsequent transition to the P-site. In the P-site, the tRNA acceptor arm is primarily in contact with the uL27m N-terminal loop. From the P-site upon peptidyl transfer, the tRNA elbow is handed over to the mL64 C-terminal helix and the deacylated tRNA is moved to the E-site, involving mL64 conformational change. The ratcheting and L1 stalk then facilitate the ejection of tRNA from the E-site.

Through the different stages of mt-tRNA movement, mL40, mL48 and mL64 undergo specific conformational changes, which support the progress of mt-tRNA from A- to E-site. In the absence of mt-tRNAs, these protein elements involved in translocation are disordered or structurally

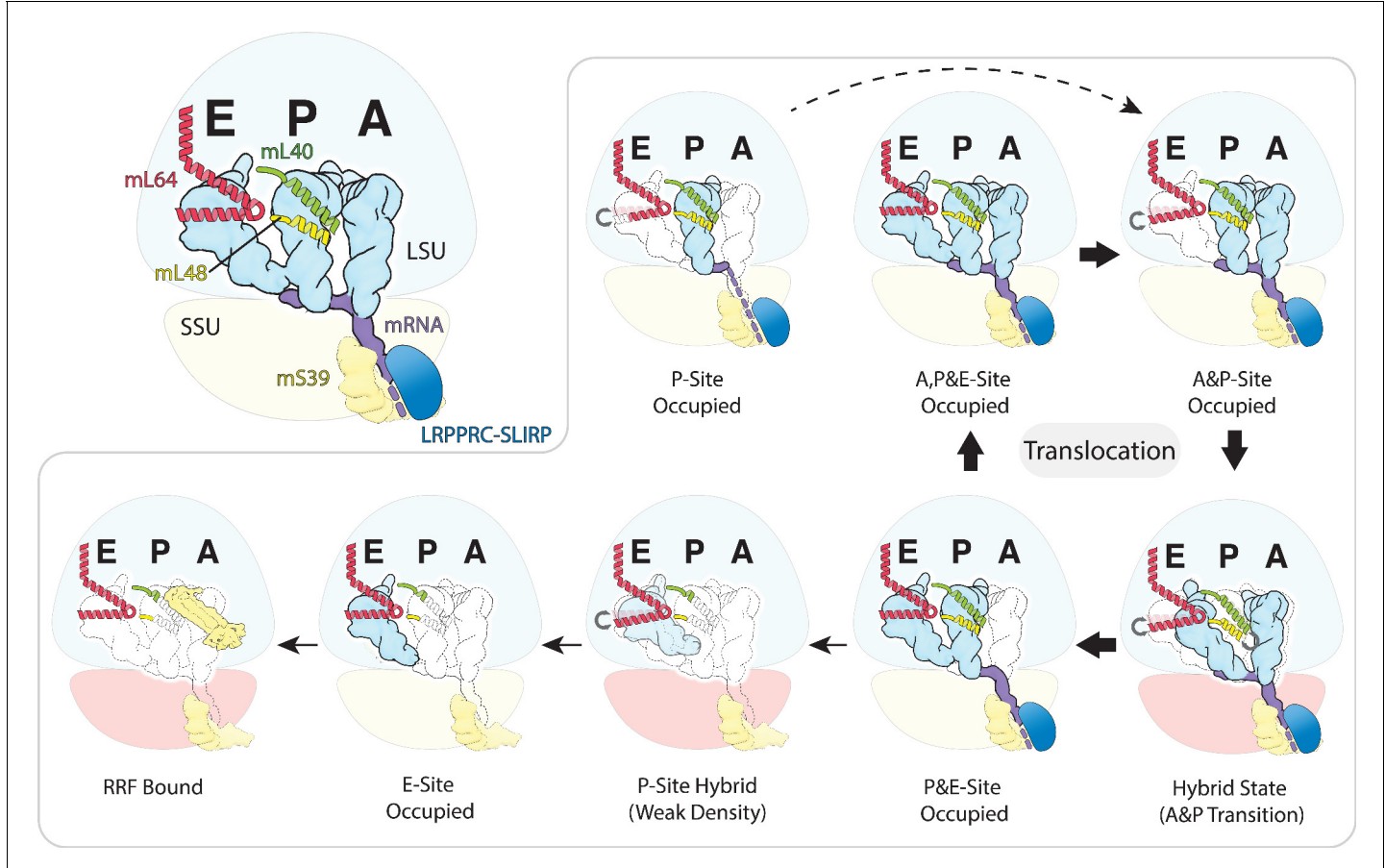

**Figure 4.** Schematic of the mt-tRNA moving through the mitoribosome based on the solved structures. Top left, mt-tRNAs bound to A-, P- and E-site and coordinated by mL40 (green) and mL64 (red), mRNA (purple) and LRPPRC-SLIRP module (blue) bound to mS39 (yellow). Seven structures with bound tRNA are arranged in the order representing tRNA translocation through the translation cycle. The conformational changes of mL40 and mL64 are indicated.

heterogenous. Therefore, at all three sites protein elements unique to the mitoribosome play roles in the tRNA recognition, particularly the elbow region. The multiple recognition events relying on the tRNA elbow and its specific coordination are likely co-evolved with the structural properties of human mt-tRNA, and they are different from fungi (*Naschberger et al., 2020*) and alveolates (*Tobiasson and Amunts, 2020*). This suggests that mitochondria-specific proteins promote effective transit through the mitoribosome by stabilizing the tRNA elbow and favoring the unidirectionality of movement.

The observation that presence of mt-tRNA and mt-mRNA on the mitoribosome is paired with the functionally associated *trans*-acting LRPPRC-SLIRP module (*Figure 4*) suggests a mechanism for delivering mt-transcripts. LRPPRC is a mt-mRNA chaperone that relaxes secondary structures (*Siira et al., 2017*). Its involvement in coordination of mitochondrial translation has been proposed (*Ruzzenente et al., 2012*), and mutations were shown to lead to Leigh Syndrome and high mortality due to episodes of severe acidosis (*Mootha et al., 2003*). The phenotype resembles that of patients with decreased levels of mitochondrial translation (*Boczonadi et al., 2018*). Therefore, our direct evidence that LRPPRC-SLIRP is engaged in delivering transcripts to the mRNA channel of the human mitoribosome is in line with the biochemical and physiological data.

Unlike in bacteria, most of the human mt-mRNAs are leaderless, and counterparts for the yeast mitochondrial translational activators system are not found (*Herrmann et al., 2013*; *Ott et al., 2016*). Therefore, how human mitoribosomes are adapted to receiving a nascent mt-mRNA was not known. In bacteria, ribosomes are directly associated with RNA polymerase, thus regulating the rate of gene expression (*Kohler et al., 2017*; *Demo et al., 2017*). In mitochondria, such coupling does

not occur because transcription and translation are most likely compartmented. However, the regulation of mitochondrial gene expression is particularly important, as it has to be coordinated with the cytosolic translation responsible for the synthesis of the complementary proteins constituting the oxidative phosphorylation system (*Couvillion et al., 2016*). Our data suggest that during translation, the PPR mitoribosomal protein mS39 located close to the mRNA channel entry acts as a platform for the binding of LRPPRC-SLIRP (*Figure 3—figure supplements 5* and *6*). It further rationalizes the previous observations that human mitoribosome dismissed the intrinsic ribosomal protein elements assisting in mRNA unfolding for active translation in bacteria such as uS4, C-terminal uS3m, and OB domains of bS1m (*Brown et al., 2014*; *Amunts et al., 2015*). Hence, the human mitoribosome evolved a specialized mechanism involving auxiliary factors and linker PPR protein mS39 for regulation and alignment of the transcripts. Our results suggest how mt-mRNA, which is transcribed at the membrane-associated nucleoid, is delivered to the human mitoribosome.

## Materials and methods

### Cell treatment and isolation of mitochondria

HEK293S-derived cells (line T501, GnTI⁻ stably expressing a transporter under tetracycline-inducible control) were grown in Freestyle 293 Expression Medium containing 5% tetracycline-free FBS in vented shaking flasks at 37°C, 5% $CO_2$ and 120 rpm. The identity has been authenticated using STR profiling. The cell lines tested negative for mycoplasma contamination. Culture was scaled up sequentially, by inoculating at $1.5 \times 10^6$ cells/mL and subsequently splitting at a cell density of $3.0 \times 10^6$ cells/mL up to a final volume of 2 L of cell culture. The cells were harvested from the 2 L culture when the cell density was $4.2 \times 10^6$ cells/mL by centrifugation at 1000 g for 7 min, 4°C (*Aibara et al., 2018*). The pellet was washed and resuspended in 200 mL Phosphate Buffered Saline (PBS). The washed cells were pelleted at 1000 g for 10 min at 4°C. The resulting pellet was resuspended in 120 mL of MIB buffer (50 mM HEPES-KOH, pH 7.5, 10 mM KCl, 1.5 mM $MgCl_2$, 1 mM EDTA, 1 mM EGTA, 1 mM DTT, protease inhibitors) and allowed to swell in the buffer for 15 min in the cold room by gentle stirring. About 45 mL of SM4 buffer (840 mM mannitol, 280 mM sucrose, 50 mM HEPES-KOH, pH 7.5, 10 mM KCl, 1.5 mM $MgCl_2$, 1 mM EDTA, 1 mM EGTA, 1 mM DTT, protease inhibitors) was added to the cells in being stirred in MIB buffer and poured into a $N_2$ cavitation device kept on ice. The cells were subjected to a pressure of 500 psi for 20 min before releasing the nitrogen from the chamber and collecting the lysate. The lysate was clarified by centrifugation at 800 g and 4°C, for 15 min, to separate the cell debris and nuclei. The supernatant was passed through a cheesecloth into a beaker kept on ice. The pellet was resuspended in half the previous volume of MIBSM buffer (three volumes MIB buffer + 1 vol SM4 buffer) and homogenized with a Teflon/glass Dounce homogenizer. After clarification as described before, the resulting lysate was pooled with the previous batch of the lysate and subjected to centrifugation at 1,000 g, 4°C for 15 min to ensure complete removal of cell debris. The clarified and filtered supernatant was centrifuged at 10,000 g and 4°C for 15 min to pellet crude mitochondria. Crude mitochondria were resuspended in 10 mL MIBSM buffer and treated with 200 Units of RNase free DNase for 20 min in the cold room to remove contaminating genomic DNA. Crude mitochondria were again recovered by centrifugation at 10,000 g, 4°C for 15 min and gently resuspended in 2 mL SEM buffer (250 mM sucrose, 20 mM HEPES-KOH, pH 7.5, 1 mM EDTA). Resuspended mitochondria were subjected to a sucrose density step-gradient (1.5 mL of 60% sucrose; 4 mL of the 32% sucrose; 1.5 mL of 23% sucrose and 1.5 mL of 15% sucrose in 20 mM HEPES-KOH, pH 7.5, 1 mM EDTA) centrifugation in a Beckmann Coulter SW40 rotor at 28,000 rpm for 60 min. Mitochondria seen as a brown band at the interface of 32% and 60% sucrose layers was collected and snap-frozen using liquid nitrogen and transferred to −80°C.

### Purification of mitoribosomes

Frozen mitochondria were transferred on ice and allowed to thaw slowly. Lysis buffer (25 mM HEPES-KOH, pH 7.5, 100 mM KCl, 10 mM MgOAc, 1.7% Polyethylene glycol octylphenyl ether, 2 mM DTT, protease inhibitors) was added to mitochondria and the tube was inverted several times to ensure mixing. A small Teflon/glass Dounce homogenizer was used to homogenize mitochondria for efficient lysis After incubation on ice for 5–10 min, the lysate was clarified by centrifugation at 30,000

g for 20 min, 4°C. The clarified lysate was carefully collected. Centrifugation was repeated to ensure complete clarification. A volume of 1 mL of the mitochondrial lysate was applied on top of 1 M sucrose in a ratio of 2.5:1. Centrifugation was carried out at 73,000 rpm for 45 min in a TLA120.2 rotor at 4°C. The pellets thus obtained were washed and sequentially resuspended in a total volume of 100 µl resuspension buffer (20 mM HEPES-KOH, pH 7.5, 100 mM KCl, 10 mM MgOAc, 1% Triton X-100, 2 mM DTT). The sample was clarified twice by centrifugation at 18,000 g for 10 min at 4°C. The sample was applied on to a linear 15–30% sucrose (20 mM HEPES-KOH, pH 7.5, 100 mM KCl, 10 mM MgOAc, 0.05% n-dodecyl-β-D-maltopyranoside, 2 mM DTT) gradient and centrifuged in a TLS55 rotor at 50,000 rpm for 120 min at 4°C. The gradient was fractionated into 50 µL volume aliquots. The absorption for each aliquot at 260 nm was measured and fractions corresponding to the monosome peak were collected. The pooled fractions were subjected to buffer exchange with the resuspension buffer to dilute away sucrose, and incubated with a bacterial antibiotic quinupristin/dalfopristin (Santa Cruz Biotechnology, 126602-89-9) at 2.2 mM for 30 min.

## Mass spectrometry

The samples for mass spectrometry analysis were prepared to identify the non-ribosomal proteins associated with the translating mitoribosome. The complex was purified exactly as for the cryo-EM analysis. The gradient was fractionated, and the monosome peak was collected. The pooled fractions were subjected to buffer exchange with the resuspension buffer to dilute away sucrose. The purified solution was submitted for mass spectrometry analysis. The data were visualized and analyzed using Scaffold_4.7.5. The proteins identified by mass spectrometry are given in *Supplementary file 2*. Apart from the mitoribosomal proteins and pyruvate dehydrogenase components, which represent a typical contamination due to the similar molecular weight of the complex, only three additional proteins were found: LRPPRC, SLIRP and mitochondrial transcription factor A. The weighted spectra were quantified, and the values for LRPPRC and SLIRP corresponded to the expected values for mitoribosomal proteins, whereas the value for the transcription factor is substantially lower.

## Electron microscopy and image processing

For cryo-EM analysis, 3 µL of ~120 nM mitoribosomes was applied onto a glow-discharged (20 mA for 30 s) holey-carbon grid (Quantifoil R2/2, copper, mesh 300) coated with continuous carbon (of ~3 nm thickness) and incubated for 30 s in a controlled environment of 100% humidity and 4°C temperature. The grids were blotted for 3 s, followed by plunge-freezing in liquid ethane, using a Vitrobot MKIV (FEI/Thermofischer). The data were collected on FEI Titan Krios (FEI/Thermofischer) transmission electron microscope operated at 300 keV, using C2 aperture of 70 µm; slit width of 15 eV on a GIF quantum energy filter (Gatan). A K2 Summit detector (Gatan) was used at a pixel size of 1.05 Å (magnification of 130,000X) with a dose of ~30 electrons/$Å^2$ fractionated over 20 frames. A defocus range of 0.8 to 2.8 µm was used. Detailed parameters are listed in *Supplementary file 1*.

Beam-induced motion correction and per-frame B-factor weighting were performed for all movies using MotionCorr2 (*Zheng et al., 2017*). Motion-corrected micrographs were used for contrast transfer function (CTF) estimation with gctf (*Zhang, 2016*). Unusable micrographs were removed by manual inspection of the micrographs and their respective calculated CTF parameters. From a total of 3481 micrographs selected, 311,655 particles were picked in RELION-2.1 (*Kimanius et al., 2016*), using reference-free followed by reference-aided particle picking procedures. Reference-free 2D classification was carried out to sort 192,970 useful particles from falsely picked objects, which were then subjected to 3D classification. 3D classes corresponding to unaligned particles and mt-LSU were discarded and 143,851 monosome particles were pooled and used for 3D Auto-refinement yielding a map with an overall resolution of 3.1 Å. Resolution was estimated using a Fourier Shell Correlation cut-off of 0.143 between the two reconstructed half maps. Masked focused refinement was performed on the small subunit and the large subunit to produce maps of 3.0 Å and 3.1 Å resolution, respectively (*Figure 1A*, *Figure 1—figure supplement 1*, *Figure 1—figure supplement 2*).

Using the consensus 3.1 Å refined map as reference, the corresponding aligned particles were subjected to 3D classification with fine angle searches which gave distinct unratcheted (100,288 particles) and ratcheted (31,366 particles) states. Partial signal subtraction was carried out using a mask covering the tRNA binding sites. Signal subtracted particles were 3D classified using a mask around

A+P-sites. These sites were chosen as they displayed nebulous density, consistent with mixed occupancy. This yielded one class with E-site only occupied (36,165 particles) and three additional classes, class one showing a weak RRF-like density; class two with A-, P-sites occupied and a partial density for E-site tRNA; and class3 with partial densities in P- and E-sites but lacking any density in the A-site. The masked 3D classification was extended to the RRF-like density for class one particles. This gave two sub-classes, based on presence (14,502 particles) or absence of RRF-like density. The sub-class lacking RRF-like density was further classified with a mask covering A- and P-sites to yield two final classes containing: A/P- and P/E state (4906 particles); and P/E state alone (19,968 particles). Class 2 and class three particles were subjected to 3D-classification using a mask around A- and P-sites, yielding four final classes: A-, P- and E-sites occupied (13,350); A- and P-sites occupied (14,475 particles); P- and E-sites occupied (24,491 particles) and P-site only occupied (6692 particles) respectively (*Figure 1—figure supplement 1*, *Figure 1—figure supplement 2*). Classes containing P-site tRNA showed presence of an additional unassigned density adjacent to mS39. Focused 3D-autorefinement was performed after pooling the particles from classes exhibiting this density, using a mask covering mS39 and the unassigned density. However, the attempts to resolve this density to a quality required for accurate model building were not successful. In the mt-SSU, all 3D classifications with signal subtracted particles were performed using a T-value of 20 and no further image alignment. The maps were subjected to modulation transfer function correction, automatic B-factor sharpening and local resolution filtering using RELION.

## Model building and refinement

Human mitoribosome structure with PDBID 3J9M (*Amunts et al., 2015*) was used as a starting template and rigid body fitted into the Cryo-EM maps using UCSF Chimera (*Pettersen et al., 2004*). Further, real space refinement of the model and building of regions absent in the previous structure were carried out in Coot v0.8.9 (*Emsley and Cowtan, 2004*; *Brown et al., 2015*). Models of mt-mRNA and mt-tRNA were built using respective models from PDBID 4V51 (*Selmer et al., 2006*) as a template and the ribonucleotides were then mutated to non-specific purine (P5P) and pyrimidine (Y5P) ribose monophosphate monomers. Human bL12m sequence information from Uniprot ID P52815 was used to build the human mitochondria-specific N-terminal extension of bL12m in coot v0.8.9. Human mt-RRF was built using PDBID 6ERI (*Perez Boerema et al., 2018*) as a template. Final models were further subjected to refinement against respective B-factor sharpened and local resolution filtered maps with Phenix.real_space_refinement v1.13_2998 (*Afonine et al., 2018*), wherein, four macro-cycles of global energy minimization with secondary structure restraints, Ramachandran and rotamer restraints were carried out for each model. Additional restraints for mt-mRNA, mt-tRNA monomers (Y5P; P5P), quinupristin (H8Q) and dalfopristin (DOL) were generated with phenix.readyset. Refined models were validated with MolProbity v.4.3.1 (*Chen et al., 2010*). Model refinement data are listed in *Supplementary file 2*.

## Structure analysis and figures

Visualization and analysis of the models and maps was carried out using UCSF Chimera (*Pettersen et al., 2004*) or PyMOL 1.8 (*DeLano, 2002*). For model and map comparisons, models were superposed in Coot v0.8.9 (*Emsley and Cowtan, 2004*; *Brown et al., 2015*) using the Secondary Structure Matching algorithm and maps downloaded from EMDB were resampled on our maps in UCSF Chimera.

## Acknowledgements

The authors thank the SciLifeLab cryo-EM and mass spectrometry facilities. This work was supported by the Swedish Foundation for Strategic Research (FFL15:0325), Ragnar Söderberg Foundation (M44/16), Swedish Research Council (NT_2015–04107), Cancerfonden (2017/1041), European Research Council (ERC-2018-StG-805230), Knut and Alice Wallenberg Foundation (2018.0080), EMBO Young Investigator Program, FEBS Long-Term Fellowship (SA), EMBO Short-Term Fellowship (AM) and H2020-MSCA-ITN-2016 (VS). The cryo-EM facility is funded by the Knut and Alice Wallenberg, Family Erling Persson, and Kempe foundations.

## Additional information

### Funding

| Funder | Grant reference number | Author |
| --- | --- | --- |
| Cancerfonden | 2017/1041 | Alexey Amunts |
| Vetenskapsrådet | NT_2015-04107 | Alexey Amunts |
| Ragnar Söderbergs stiftelse | M44/16 | Alexey Amunts |
| Horizon 2020 Framework Programme | ERC-2018-StG-805230 | Alexey Amunts |
| Knut och Alice Wallenbergs Stiftelse | 2018.0080 | Alexey Amunts |
| Swedish Foundation for Strategic Research | FFL15:0325 | Alexey Amunts |
| EMBO | EMBO Young Investigator Program | Alexey Amunts |
| Federation of European Biochemical Societies | FEBS Long-Term Fellowship | Shintaro Aibara |
| EMBO | EMBO Short-Term Fellowship | Angelika Modelska |
| Horizon 2020 - Marie Sklodowska-Curie Innovative Training Network | 721757 | Vivek Singh |

The funders had no role in study design, data collection and interpretation, or the decision to submit the work for publication.

### Author contributions

Shintaro Aibara, Conceptualization, Data curation, Formal analysis, Validation, Investigation, Visualization, Writing - original draft, Writing - review and editing; Vivek Singh, Validation, Investigation, Visualization, Writing - original draft, Writing - review and editing; Angelika Modelska, Conceptualization; Alexey Amunts, Conceptualization, Resources, Supervision, Funding acquisition, Investigation, Methodology, Writing - original draft, Project administration, Writing - review and editing

### Author ORCIDs

Vivek Singh (iD) https://orcid.org/0000-0003-4656-3362
Alexey Amunts (iD) https://orcid.org/0000-0002-5302-1740

### Decision letter and Author response

Decision letter https://doi.org/10.7554/eLife.58362.sa1
Author response https://doi.org/10.7554/eLife.58362.sa2

## Additional files

### Supplementary files

• Supplementary file 1. Cryo-EM data collection parameters.

• Supplementary file 2. The list of proteins identified by mass spectrometry with their accession numbers and molecular weight. The proteins are listed according to the weighted spectra from high to low that was quantified using Scaffold_4.7.5. The pyruvate dehydrogenase complex components and non-mitoribosomal components are shown in italic; LRPPRC and SLIRP are highlighted in yellow.

• Supplementary file 3. Model refinement and validation statistics. * FSC corrected for the effect of the mask according to 0.143-cutoff criterion ** FSC (masked) according to 0.5-cutoff criterion

• Supplementary file 4. The amino acid sequence for LRPPRC and SLIRP proteins is displayed. The peptides identified by mass spectrometry that meet 40% minimum threshold are highlighted in

yellow. Above the sequence, the protein accession number, molecular weight and protein name are shown together with the number of unique peptides, spectra and % coverage.

• Transparent reporting form

## Data availability

The electron density maps have been deposited in EMDB under accession codes EMD-11397, EMD-11391, EMD-11392, EMD-11394, EMD-11393, EMD-11395, EMD-11396 and EMD-11390. All models have been deposited in PDB under accession codes 6ZSG, 6ZSA, 6ZSB, 6ZSD, 6ZSC, 6ZSE, 6ZSF and 6ZS9. All data is available in the paper or Supplementary Information.

The following datasets were generated:

| Author(s) | Year | Dataset title | Dataset URL | Database and Identifier |
|---|---|---|---|---|
| Aibara S, Singh V, Amunts A | 2020 | Human mitoribosome in complex with mRNA and A-site tRNA, P-site tRNA, E-site tRNA | https://www.ebi.ac.uk/pdbe/entry/emdb/EMD-11397 | Electron Microscopy Data Bank, EMD-11397 |
| Aibara S, Singh V, Amunts A | 2020 | Human mitoribosome in complex with mRNA and A-site tRNA, P-site tRNA | https://www.ebi.ac.uk/pdbe/entry/emdb/EMD-11391 | Electron Microscopy Data Bank, EMD-11391 |
| Aibara S, Singh V, Amunts A | 2020 | Human mitoribosome in complex with mRNA and P-site tRNA | https://www.ebi.ac.uk/pdbe/entry/emdb/EMD-11392 | Electron Microscopy Data Bank, EMD-11392 |
| Aibara S, Singh V, Amunts A | 2020 | Human mitoribosome in complex with mRNA and P-site tRNA, E-site tRNA | https://www.ebi.ac.uk/pdbe/entry/emdb/EMD-11394 | Electron Microscopy Data Bank, EMD-11394 |
| Aibara S, Singh V, Amunts A | 2020 | Human mitoribosome in complex with E-site tRNA | https://www.ebi.ac.uk/pdbe/entry/emdb/EMD-11393 | Electron Microscopy Data Bank, EMD-11393 |
| Aibara S, Singh V, Amunts A | 2020 | Human mitoribosome in complex with mRNA and A/P-tRNA, P/E-tRNA | https://www.ebi.ac.uk/pdbe/entry/emdb/EMD-11395 | Electron Microscopy Data Bank, EMD-11395 |
| Aibara S, Singh V, Amunts A | 2020 | Human mitoribosome in complex with mRNA and P/E-tRNA | https://www.ebi.ac.uk/pdbe/entry/emdb/EMD-11396 | Electron Microscopy Data Bank, EMD-11396 |
| Aibara S, Singh V, Amunts A | 2020 | Human mitoribosome in complex with recycling factor | https://www.ebi.ac.uk/pdbe/entry/emdb/EMD-11390 | Electron Microscopy Data Bank, EMD-11390 |
| Aibara S, Singh V, Amunts A | 2020 | Human mitoribosome in complex with mRNA and A-site tRNA, P-site tRNA, E-site tRNA | https://www.rcsb.org/structure/6ZSG | RCSB Protein Data Bank, 6ZSG |
| Aibara S, Singh V, Amunts A | 2020 | Human mitoribosome in complex with mRNA and A-site tRNA, P-site tRNA | https://www.rcsb.org/structure/6ZSA | RCSB Protein Data Bank, 6ZSA |
| Aibara S, Singh V, Amunts A | 2020 | Human mitoribosome in complex with mRNA and P-site tRNA | https://www.rcsb.org/structure/6ZSB | RCSB Protein Data Bank, 6ZSB |
| Aibara S, Singh V, Amunts A | 2020 | Human mitoribosome in complex with mRNA and P-site tRNA, E-site tRNA | https://www.rcsb.org/structure/6ZSD | RCSB Protein Data Bank, 6ZSD |
| Aibara S, Singh V, Amunts A | 2020 | Human mitoribosome in complex with E-site tRNA | https://www.rcsb.org/structure/6ZSC | RCSB Protein Data Bank, 6ZSC |
| Aibara S, Singh V, Amunts A | 2020 | Human mitoribosome in complex with mRNA, A/P-tRNA, P/E-tRNA | https://www.rcsb.org/structure/6ZSE | RCSB Protein Data Bank, 6ZSE |
| Aibara S, Singh V, Amunts A | 2020 | Human mitoribosome in complex with mRNA and P/E-tRNA | https://www.rcsb.org/structure/6ZSF | RCSB Protein Data Bank, 6ZSF |
| Aibara S, Singh V, Amunts A | 2020 | Human mitoribosome in complex with recycling factor | https://www.rcsb.org/structure/6ZS9 | RCSB Protein Data Bank, 6ZS9 |

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
