## [Decision Letter]

**Acceptance summary:**

Aibara et al. illuminate how gene expression within the mitochondrial matrix is governed by the human mitochondrial ribosome. Through analysis of eight new structures of different compositional and conformation mito-ribosome states, their manuscript teaches us how mito-specific ribosomal proteins guide tRNA movements through the mito-ribosome. Their structures also suggest how a novel complex, a heterodimer of the proteins LRPPRC-SLIRP, engages the mRNA entry site in the small ribosomal subunit to facilitate translation, providing a potential explanation for how mutations in LRPPRC causes Leigh syndrome.

**Decision letter after peer review:**

Thank you for submitting your article "Structural basis of mitochondrial translation" for consideration by *eLife*. Your article has been reviewed by three peer reviewers, and the evaluation has been overseen by a Reviewing Editor and Cynthia Wolberger as the Senior Editor. Three of the reviewers have opted to remain anonymous.

The reviewers have discussed the reviews with one another and the Reviewing Editor has drafted this decision to help you prepare a revised submission.

Summary:

Aibara et al. report improved resolution updates of the human mitoribosome structure with matching insights into aspects of translation within the mitochondrial matrix. Their cryo-EM reconstructions of mito-ribosomes purified in the presence of antibiotics revealed eight different compositional or conformational states. Analysis of these states suggests new roles for a subset of the 36 mito-specific ribosome proteins in the mechanics of translation-including guiding mt-tRNA movements through the ribosome and the docking of leaderless mt-mRNA to the small subunit. Of particular note, the authors discovered a new density bound to the mitochondria-specific protein mS39 that correlated with well-resolved mRNA density. Mass spectrometry of the purified sample used for cryoEM suggests this density is attributable to a heterodimer of LRPPRC-SLIRP. While the cryo-EM density for this complex is too blurry to interpret, the notion that these proteins "hand-off" messages to the small mito-ribosomal subunit is attractive and consistent with prior data showing that these two proteins play roles in transcription-translation coordination and that mutations in LRPPRC are associated with Leigh syndrome. Aibara also present analyses of proteins mL40, mL48, and mL64-plus a density that may correspond with polyamines like spermine-during the translocation of mt-tRNAs. Finally, the authors provide improved models of other functional regions, such as the L7/L12 stalk. Overall, this manuscript is an exciting advance in understanding the structural basis of translation by the highly divergent mitoribosome and tRNAs-but specific outstanding issues should be addressed before acceptance.

Essential revisions:

1) The authors write, "the structures of human mt-tRNAs are remarkably diverse, including non-canonical and truncated species with reduced D- and/or T-loops." From the structural data presented here, though, which must be an average of many different tRNA species, it appears the authors modeled tRNA using generic structures that are indistinguishable, at this resolution, from elbow-shaped cytosolic tRNAs. Please described whether and how the interactions seen between tRNA and mL40 or mL48 depend on structural properties of mito-tRNAs. If so, please support this claim by showing the cryo-EM maps used to support the modeling of the tRNA "elbow" regions and contacts with mL40 and mL64.

2) The interpretation of the density between the A-site and P-site tRNAs as a polyamine is reasonable but speculative-it could be protein. The authors acknowledge this in the Results, yet reach too far by highlighting polyamines in the Abstract and Discussion. Was the putative spermine density found bound in a position similar to the first crystal structure of tRNA-Phe? Is the alleged polyamine density present in all the maps or only those containing tRNAs in the P and A sites? Please discuss and mention which other polyamines are present in mitochondria that could, in principle, correspond with the density attributed to spermine here.

3) Given the poor resolution of the mS39-adjacent density (Figure 3—figure supplement 5), the authors should soften their claim to have unambiguously "identified" this as LRPPRC-SLIRP. Based on the mass spectrometry and EM data, this is certainly a reasonable hypothesis worthy of discussion. Still, the authors are encouraged to be more cautious in their interpretation of the density, and the figures and text modified accordingly.

---

## [Author Response]

Essential revisions:1) The authors write, "the structures of human mt-tRNAs are remarkably diverse, including non-canonical and truncated species with reduced D- and/or T-loops." From the structural data presented here, though, which must be an average of many different tRNA species, it appears the authors modeled tRNA using generic structures that are indistinguishable, at this resolution, from elbow-shaped cytosolic tRNAs. Please described whether and how the interactions seen between tRNA and mL40 or mL48 depend on structural properties of mito-tRNAs. If so, please support this claim by showing the cryo-EM maps used to support the modeling of the tRNA "elbow" regions and contacts with mL40 and mL64.

As requested, we have added this information in Figure 3—figure supplement 3 showing the experimental cryo-EM density for the protein elements involved in tRNA interactions. We have also included a schematic indicating the residues involved in the binding of A-, P- and E-bound tRNAs. This representation clearly shows that the interactions are concentrated at the elbow regions for A- and P-site, and the elbow remains stabilized by the mitochondria-specific protein elements throughout the moving, consistent with the text. Since the density for tRNA is produced from an average of many different tRNA identities, we refrain from commenting on the exact tRNA nucleotide involved in the interaction and limit the interpretation to sections (elbow, anti-codon stem, acceptor stem), as the reviewers comment correctly.

2) The interpretation of the density between the A-site and P-site tRNAs as a polyamine is reasonable but speculative-it could be protein. The authors acknowledge this in the Results, yet reach too far by highlighting polyamines in the Abstract and Discussion. Was the putative spermine density found bound in a position similar to the first crystal structure of tRNA-Phe? Is the alleged polyamine density present in all the maps or only those containing tRNAs in the P and A sites? Please discuss and mention which other polyamines are present in mitochondria that could, in principle, correspond with the density attributed to spermine here.

As requested, we have removed the too far reaching section from the Discussion. Since the density present only in the maps containing tRNAs in the P and A sites, it would be too speculative to develop further discussion, and our structural data is poor in this region and does not provide any functional insight with to respect the mechanism. Therefore, this part of the manuscript has been toned down, and the following changes were made:

Abstract: deleted “polyamine”;

Replaced “we modeled it as spermine and refined the structures” with “it could be modeled it as spermine”;

Removed the spermine from the coordinates;

Removed the model for spermine from Figure 3—figure supplement 2 and corrected the legend accordingly.

3) Given the poor resolution of the mS39-adjacent density (Figure 3—figure supplement 5), the authors should soften their claim to have unambiguously "identified" this as LRPPRC-SLIRP. Based on the mass spectrometry and EM data, this is certainly a reasonable hypothesis worthy of discussion. Still, the authors are encouraged to be more cautious in their interpretation of the density, and the figures and text modified accordingly.

As requested, we revised the relevant sections.

Deleted “to identify this component of the mitoribosome:mRNA:tRNA-associated complex”.

Results: replaced “revealed” with “suggested”. and deleted “a single possibility”.

Figure 3—figure supplement 5: added “density”.